# Carbon Footprint Research Based on Input–Output Model—A Global Scientometric Visualization Analysis

**DOI:** 10.3390/ijerph191811343

**Published:** 2022-09-09

**Authors:** Jingwei Han, Zhixiong Tan, Maozhi Chen, Liang Zhao, Ling Yang, Siying Chen

**Affiliations:** 1School of Economics and Business Administration, Chongqing University, Chongqing 400044, China; 2School of Public Policy and Administration, Chongqing University, Chongqing 400044, China; 3School of Economics and Business Administration, Chongqing University of Education, Chongqing 400067, China; 4Institute of Scientific Research and Development, Chongqing University, Chongqing 400044, China

**Keywords:** carbon footprint, input–output model, knowledge-mapping analysis, visual analysis

## Abstract

Reducing the effect of mankind’s activities on the climate and improving adaptability to global warming have become urgent matters. The carbon footprint (CF), derived from the concept of ecological footprint, has been used to assess the threat of climate change in recent years. As a “top to bottom” method, input–output analysis (IOA) has become a universally applicable CF assessment tool for tracing the carbon footprint embodied in economic activities. A wide range of CF studies from the perspective of the IOA model have been presented and have made great progress. It is crucial to have a better understanding of what the relevant research focuses on in this field, yet so far a systematic synopsis of the literature is missing. The purpose of this paper is to explore the knowledge structure and frontier trends in respect of the IOA model applied to CF research using scientometric visualization analysis. The main findings of this paper are as follows. (1) Published articles show a two-stage increase in the period 2008 to 2021, and present a complex academic network of countries, authors, and institutions in this important domain. (2) The classic studies are mainly divided into three categories: literature reviews, database application introduction, and CF accounting in different scales. (3) The research hotspots and trends show that the research scales tend to be more microscopic and applications of models tend to be more detailed. In addition, supply-chain analysis and driver-factor analysis will probably become the main research directions in the future.

## 1. Introduction

With the development of industrialization, global warming has caused irreversible damage to the environment that human beings depend on in terms of sea level rise, food crises, water shortages and so on. Scientific research shows that greenhouse gas (GHG) emissions are the main cause of global warming. Reducing GHG emissions has become the worldwide target since the first Intergovernmental Panel on Climate Change (IPCC) assessment report was released in 1990 [1]. In 2014, the IPCC issued its fifth assessment report and stated that global average land temperature had increased by 0.85 °C from 1880 to 2012 [2]. To avoid serious climate change impacts, the Paris Agreement proposed global warming should be limited to 1.5 °C in 2016 [3]. In 2022, the IPCC released the working group II contribution to the sixth assessment report, which indicated that even for the very low GHG emissions scenario, there is at least a greater than 50% likelihood that global warming will reach or exceed 1.5 °C in the near term [4]. The ambitious emission-reduction target requires the joint efforts of all governments and industries. In order to addressing climate change effectively, it is urgent for all countries in this world to reach a further consensus on specific implementation. Accurate assessment of GHG emissions from human activities is an essential prerequisite for carbon-reduction strategies.

The carbon footprint (CF) captures a lot of interest among many scholars. Derived from the concept of the ecological footprint [5,6], CF has become a catchphrase in public discussion. CF is a type of ecological footprint in terms of carbon emissions from individual or mass production, consumption, and organizational activities [7]. It is important to realize that CF originates from the ecological footprint but is not equivalent to it. Despite its name, CF is not expressed in terms of area, as the ecological footprint is [8]. The widely accepted definition of CF is the total amount of carbon emissions caused by an activity directly and indirectly or accumulated of a product over its life stages, which is expressed in terms of carbon dioxide (CO_2_) equivalents generally [9]. It is widely accepted that life-cycle assessment (LCA) and input–output analysis (IOA) are useful tools for calculating CF. LCA focuses on specific products and services, while IOA focuses on product groups that typically cover everything consumed in an economy [10]. Compared to LCA, IOA can transform the complex economic relationship between production sectors or regions into the physical relationship of GHG emissions. With the advantages of clear principles and process, IOA can reflect the exchange process of emissions, thus making the direct and indirect emission relationship clear [11,12]. Judging from the soaring number of publications over the last few years, IOA has become the universally applicable assessment methods for tracing the CF embodied in economic activities. As a “top to bottom” analysis method, the input–output model (IOM) was formulated in 1936 [13] and enriched in the 1970s [14,15]. IOA provides an appropriate methodological framework to quantificationally reflect the technical and economic links among the various departments at the national and supranational level.

Recent research shows that IOA has been extensively used and applied to investigate the issues of CF at global [16,17,18], national [19,20,21], regional [22,23,24], and city [25,26,27] levels, which reflects the direct and indirect relationship between different departments comprehensively in the economic system. With the research being relatively mature and complete, recent literature has reviewed CF assessment with IOA from various perspectives, such as urban CF, spatial consumption-based CF, tourism CF, and urban ecosystems CF [28,29,30,31]. However, most literature is limited to some specific aspects instead of more systematic analysis, and there is a lack of in-depth discussion on relationships in the existing research. Bibliometric analysis can offer an overall view of visual network relationships supported by a large amount of information from the literature. To date, there have been some review articles that have adopted a scientometric visualization analysis to explore the intellectual structure and evolution history of related fields. Several examples of bibliometric applications include CF research [32,33], CF research from an LCA perspective [11], energy and environment research from an IOA perspective [34], and other related topics, such as carbon neutralization goals [35,36]. However, no studies have reported the application of IOA in the field of CF from a bibliometric perspective.

This paper tries to use the bibliometric method to explore the knowledge-evolution process of IOA quantitatively and qualitatively for CF, according to the visualization tools VOSviewer and CiteSpace. The main contributions of this paper are: (1) summarizing in detail the knowledge base and thematic development of IOA for CF; (2) systematically combing the academic cooperation network of authors, countries (regions), and institutions in this important domain; and (3) providing a better understanding of worldwide research hotspots, emerging ideas, and research trends in related research. The rest of this article is structured as follows. Section 2 presents the research design and data retrieval process. Section 3 provides the knowledge map of IOA for CF based on basic feature analysis, co-citation network analysis, structure variation analysis, and co-occurrence network analysis. Section 4 gives the conclusions.

## 2. Methods and Materials

### 2.1. Bibliometric Analysis

As a research method for exploring the knowledge structure and frontier trends of the discipline, bibliometric analysis has been widely adopted in literature review. A bibliometric analysis can present the general characteristics of a scientific issue, including authors, journals, institutes, keywords, etc. [37]. VOSviewer and CiteSpace are the commonly used analysis software, and allow a comparative evaluation of the literature. VOSviewer was developed by van Eck and Waltman in Leiden, The Netherlands [38], and can generate a variety of maps based on bibliometric relationships, such as collaboration between authors, institutions, and countries (regions), which is helpful to identify major research groups and reveal the development path of a certain discipline [39]. CiteSpace, developed by Chen Chaomei in Philadelphia, PA, USA, was designed for progressive knowledge domain visualization [40,41,42]. CiteSpace has provided an effective way for big-data measurement analysis of literature and become an emerging bibliometric analysis tool. In order to deepen the understanding of theoretical development of IOM for CF, this paper uses VOSviewer v1.6.17 and CiteSpace v5.8.R3 to carry out a bibliometric analysis. The operation process of bibliometric analysis includes keyword determination, data collection, parameter selection, visualization knowledge mapping, etc. An outline of research design is shown in Figure 1. Firstly, we described the data collection process, including keyword determination and data processing. Secondly, we conducted the basic feature analysis, including total number trend analysis of annual publications, collaboration analyses of authors, country (region) and institution by VOSviewer software, and dual-map overlay analysis between cited journals and citing journals by Citespace software. Thirdly, we conducted the knowledge structure and frontier trend analysis by Citespace software, including co-citation network analysis of cited articles, structure variation analysis of citations trajectories, and co-occurrence network analysis of keywords. Finally, the key findings are summarized and limitations discussed.

### 2.2. Data Collection

Accurate data on the literature is the key to bibliometric analysis. Therefore, to collect more comprehensive papers related to IOA for CF, this paper conducted advanced retrieval through Web of Science (WOS), which is considered a reliable database for visual analysis. Furthermore, the Web of Science Core Collection (WOSCC), including the Social Sciences Citation Index (SSCI) and the Science Citation Index Expanded (SCI-E) databases, was chosen to increase the representativeness and accessibility of the data. After repeated comparisons of multiple sets of literature data to eliminate false and missed detections, the search codes TS = (“carbon footprint *”) and TS = (“input-output” or “IOA”) were determined. The selected type was “article.” Finally, a total of 491 publications were retrieved. These covered the period from 2008 to 2021 and included all bibliographic information, such as title, author, abstract, keywords, source publications, and references.

## 3. Results and Discussion

### 3.1. Basic Feature Analysis

#### 3.1.1. Trends in the Total Number of Papers

In general, the number of annual publications reflects the importance and attention of a research field. The academic interest in IOA for CF has increased substantially over the years. As Figure 2 shows, the curve of papers published over time can be roughly divided into two stages. (1) The first stage is from 2008 to 2013, with an average annual number of 13 publications. The first publication retrieved was from 2008. After a surge in 2009, the number remained stable, forming a platform until 2013. (2) The second stage is from 2014 to 2021, with an average annual number of 52 publications. There was a significant growth in publications in 2014, nearly double that of the previous year. From then on, the numbers show a rapidly increasing trend. A possible reason is that the UN Climate Change Conference held in Paris in 2015 attracted extensive attention of the media, public, and academia on such topics as carbon emission allocation and carbon emission cooperation. Thus, it also potentially led to a marked increase in research on CF assessment using an IOA approach.

#### 3.1.2. Author, Country (Region), and Institution

Information of authors, countries (regions), and institutions for the retrieved papers are shown in Table 1, Table 2 and Table 3. The collaboration networks between author, country (region), and institution were produced by VOSviewer and displayed in Figure 3, Figure 4 and Figure 5 respectively. According to the author information listed in Table 1, Wood Richard, Lenzen Manfred, Wiedmann Thomas, Hertwich Edgar G, and Kucukvar Murat are the most representative scholars in this field. Wood Richard ranks first with 36 papers, followed by Lenzen Manfred and Wiedmann Thomas, with a total of 29 papers and 26 papers, respectively. In addition, the visualization of scientific collaboration networks shown in Figure 3 suggests the close relationships between these high-yield authors. According to the country (region) and institution information, the geographic coverage is fairly wide overall. Scholars from 51 countries (regions) carried out related research. The top 10 countries (regions) are listed in Table 2 and their collaboration network is displayed in Figure 4. China has published 146 papers, followed by the USA and Australia, with 108 and 93 papers, respectively. In terms of institutions, the Norwegian University of Science Technology ranks first with 55 papers. Leiden University ranks second with 54 papers, followed by the University of Sydney with 51 papers. Those productive institutions are listed in Table 3 and their collaboration network is shown in Figure 5.

#### 3.1.3. Dual-Map Overlays

The literature on CF assessment with IOA is distributed across multiple scientific fields, including environmental sciences (319 articles, 65.0%); environmental engineering (165 articles, 33.6%); green and sustainable science and technology (154 articles, 31.4%); environmental studies (96 articles, 19.6%); economics (88 articles, 17.9%); energy fuels (47 articles, 9.6%); ecology (33 articles, 6.7%); chemical engineering (28 articles, 5.7%); meteorology and atmospheric sciences (24 articles, 4.9%), and multidisciplinary geosciences (13 articles, 2.6%). Furthermore, through dual-map overlay analysis, this paper discusses the distribution of CF assessment with the IOA approach in the whole scientific field map. With the function of “JCR journal maps” in CiteSpace software, dual-map overlays were drawn, representing the entire dataset in the context of a global map of science generated from over 10,000 journals indexed in the WOS [43].

Dual-map overlays clearly and intuitively show the citation relationship and the process of knowledge flow between different journals by analyzing the connection and diffusion direction of knowledge in the research field. These consist of two base maps: citing journals on the left and cited journals on the right. On the left base map of citing journals, the vertical axis of the ellipse represents the number of publications, and the horizontal axis represents the number of authors. On the right cited journals, the vertical and horizontal axes represent the number of the citations of publications and the number of cited authors, respectively [44]. The links depict the citation relationship between the citing journals and citied journals. Due to the large number of journals, it is difficult to identify the knowledge-flow path in the original map intuitively. Therefore, it is necessary to use the Z-score (Z) algorithm to merge the paths. Figure 6 shows the result after Z-score processing (top) and detail in enlarged scale (bottom).

From the result of Z-score analysis, there are three main knowledge-flow paths of the journals. The knowledge carriers of CF assessment with IOA are distributed among the journal clusters on the left, and the knowledge source is distributed among the journal clusters on the right. The main knowledge-flow paths shows that 2#environment, toxicology, nutrition (Z-score = 4.2892904, f = 11,340) and 12#economics, economic, political (Z-score = 5.449707, f = 14,084) are the important knowledge-based journals of 7#veterinary animal science. Similarly, 12# economics, economic, political (Z-score = 3.732764, f = 10,024) are the important knowledge-based journals of 10#economics, economic, political. The knowledge flow can be considered as direct evidence that the research on CF assessment with IOA became diversified and interdisciplinary gradually. Moreover, representative journals are identified on the left of the dual-map overlays, such as the *Journal of Cleaner Production* (J CLEAN PROD), *Journal of Industrial Ecology* (J IND ECOL), *Economic Systems Research* (ECON SYST RES), and *Ecological Economics* (ECOL ECON). On the right of the dual-map overlays, the representative journals include the *Journal of Cleaner Production* (J CLEAN PROD), *Environmental Science and Technology* (ENVIRON SCI TECHNOL), *Ecological Economics* (ECOL ECON), and *Energy Policy* (ENERG POLICY).

### 3.2. Co-Citation Network Based on the Focus Topic

In this part, the key nodes of the cited articles in the co-citation network by CiteSpace are analyzed to identify classic papers and their core research fields. A co-citation network, based on the abundant data source of cited references, can provide wider and deeper knowledge related to the research domain [39]. Co-citation refers to the relationship due to the synchronized appearance of two cited articles in the citing article. Co-citation analysis can help researchers explore the hot-spot distributions of the specific topics [41]. In a co-citation network, clusters are connected generally by the important “bridge node,” which has relatively high citation frequency, citation bursts, and betweenness centrality. These “bridge nodes” play important roles to the transition from one period to another in the network [40], and indicate the milestones in the field of scientific research.

Citation frequency refers to the number of times the cited article appears in the citing article. In the co-citation network, the node size is positively correlated with the citation frequency. The larger the node size is, the higher the citation frequency is, and therefore the bigger influence the reference has. Citation bursts refer to the articles that have suddenly emerged or significantly increased within a specific time interval, and can reflect the dynamics of the research field. It is detected by an algorithm proposed by Kleinberg [45]. Citation bursts are an important tool for literature content mining to identify active or frontier research nodes. Those major milestones generally have strong citation bursts in science-mapping development [42]. Higher burst scores have great significance in structural breakthroughs. The importance of a node depends on the indicator betweenness centrality in the co-citation network. The definition of betweenness centrality is shown in Equation (1): *ρ_jk_* is the shortest paths’ number between node *j* and node *k*, and *ρ_jk_* (*i*) is the number of those paths that pass through node *i*. [46,47].
(1)$$\mathrm{Centrality}\;\left(\mathrm{node}\;i\right)=\sum_{i\neq j\neq k}\frac{\rho_{jk}\;\left(i\right)}{\rho_{jk}}$$

To ensure the rigor of the results, a series of parameters in CiteSpace were set: data collection period = 2008 to 2021, year slice = 1, node types = cited reference, link strength = cosine, link scope = with slices, data selection criterion = modified g-index (the scale factor *k* is 25). The co-citation network consisting of a total of 675 nodes and 2168 links is shown in Figure 7. Each node represents a cited reference, named by author and publication year. Figure 7 also shows the top 15 articles with the strongest citation bursts.

To comprehensively find the classic literature that constitutes the knowledge basis of this research field, another ten cited articles were selected with high frequency and betweenness centrality (Table 4). After reading and combing the literature in detail, we divided these important node references into three main areas: related literature reviews, database application introduction, and CF accounting in different scales, which will be discussed in the following three sections.

#### 3.2.1. Key Network Nodes of Related Literature Reviews

A systematic review article can describe the basic outline of the research field and provide a quicker understanding for other scholars. There are several review articles that are key network nodes with high parameter value in the co-citation network. Wiedmann and Minx [9] provided a systematic literature review in environmental IOA for CF, and summarized the applications of single-region input–output model (SRIO), multiregion input–output model (MRIO), and so on. After that, Wiedmann [50] provided a systematic review of the MRIO for consumption-based environmental accounting and concluded that improvements in data quality and model accuracy were needed in further research. Minx et al. [49] provided an overview of IOA applications for CF accounting using evidence from the UK, which made an additional contribution to this emerging field. Tukker and Dietzenbacher [53] introduced systematic research on a global multiregional input–output model (GMRIO) and provided a short historical context of GMRIO. Wiedmann and Lezen [56] reviewed articles focusing on environmental and social footprints of global trade based on the GMRIO model, which made scientific advances in the following four areas: new indicators, modeling impacts, spatial resolution, and collaboration. Overall, those review articles played an important role across different disciplines in the co-citation network, which are the basic knowledge framework in the research domain of CF accounting based on IOA.

#### 3.2.2. Key Network Nodes of Database Application Introduction

Measuring CF with IOM requires appropriate frameworks and databases. Due to the absence of credible data, early empirical applications mainly focused on single countries with SRIO. As global environmental problems grow rapidly, the increasing demand for MRIO has led to several large research projects using existing collections of national input–output tables (IOTs) to construct detailed and accurate MRIO tables. These global MRIO databases are the basic of CF analysis. There are several key network nodes of articles in the co-citation network that separately introduced the main widely applied MRIO databases, such as Exiobase (Exiopol Database), EORA, WIOD, and GTAP.

##### Exiobase

Exiobase combines national supply–use tables (SUTs) and IOTs with international trade and energy and resource-extraction data, and is one of the most suitable databases for CF study. In 2009, Tukker et al. [58] introduced the EU-funded project Exiopol. The project aimed to develop an environmentally extended IOA framework for EU 27 countries to evaluate environmental impacts of different economic sectors’ activities expressed as external costs, ecological footprints, and so on. Exiopol overcame the significant limitations of data sources of multiregional environmentally extended SUTs and IOTs [59]. Wood et al. [55] explored the methods to arrive at the Exiobase. The major strength of Exiobase is that it can provide more sector details, which contributes to a more detailed insight into the impacts of production and consumption. Compared with other MRIO datasets, Exiobase includes more social and environmental data for calculating CF in most cases. Stadler et al. [57] described Exiobase3, the latest version built upon the previous one. Exiobase3 uses rectangular SUTs in 163 industries by 200 product classifications of 44 countries ranging from 1995 to 2011, which has become a unique tool to analyze the environmental pressures generated by economic activities over time.

##### EORA

EORA is a popular input–output database commonly used by scholars that contains 189 countries or regions. Lenzen et al. [52,60] presented the new MRIO database EORA funded by the Australian Research Council (ARC). Due to the high level of automation, standardization, and organization, the project is continuously updating (about 2 years lag). At present, the EORA database is available as a long continuous time series of MRIO tables spanning from 1990 to 2021, which enables identification of the key drivers and trends of global climate change robustly. The EORA database disaggregated the world into countries and sectors in as much detail as possible and refrained from changing the original raw data structure, and thus it is widely applied to accurate footprint-type assessments of global trade.

##### WIOD

With high data accuracy, WIOD has also become one of the most widely used data-base in CF studies. Dietzenbacher et al. [61] discussed the project of the World Input–Output Database (WIOD) and the compilation of world input–output tables (WIOTs). The research described how information has been harmonized and reconciled to obtain a consistent time series of WIOTs from international trade statistics, national accounts statistics, and supply–use tables. The WIOD database distinguishes 35 industries and 59 products of 40 countries, and contains a time series from 1995 to 2011. The WIOD project aimed to create an all-encompassing database to address the quest for both policymakers and academic researchers, and consequently led to more empirical studies examining the global socioeconomic and environmental impact.

##### GTAP

GTAP contains detailed accounts of production, consumption, bilateral trade, and data on CO_2_ emissions. Peters et al. [62] demonstrated how to use the Global Trade Analysis Project (GTAP) database to build an MRIO table. In the 1990s, the GTAP was implemented, aiming to set up a database consisting of bilateral trade data, IOTs, and other important economic data. The GTAP database has excellent regional detail and reasonable sector detail without additional balancing for the MRIO table, because it is already balanced. While there are some advantages or disadvantages, the GTAP database remains one of the most suitable resources for MRIO models in the short term.

#### 3.2.3. Key Network Nodes of CF Accounting

The input–output model, originating from economics, is applied in the field of embodied carbon due to the advantages of coping with the complicated network data structures [63]. There are several key nodes of empirical studies and cases studies on CF accounting in the co-citation network. Early scholars paid more attention to the shared responsibility of carbon emission and the embodied carbon in international trade. Peters [64] used a generalized environmental IOA to determine the embodied carbon emissions embodied in trade (EET). The result shows that globally there are considerable flows of anthropogenic carbon embodied in international trade and those countries in the Annex B of Kyoto Protocol are the main net importers. Due to the limited participation in binding commitments, the carbon leakage through EET can significantly undermine the efforts of global climate policy. Furthermore, to address the issues of excluding international transportation and potential carbon leakage under the United National Framework Convention of Climate Change, Peters [65] discussed a consistent method of weighting production-based and consumption-based national emission inventories, which is an extension of the previous literature. Peters [62] used both bilateral trade and MRIO models to identify emission transfers via international trade. The results indicated that from the production or consumption perspective, international trade can explain the emissions change in many countries. Most developed countries maintained stable emissions partially caused by growing imports from developing countries. Given that emission transfers are becoming more and more significant via international trade, the climate polies should not be separated from the trade policies. Hertwich and Peters [48] analyzed the contribution of eight consumption categories to GHG emissions, providing a different perspective on the drivers at a global level. Davis and Caldeira [66] conducted global consumption-based carbon emission accounting with a fully coupled MRIO model and provided evidence that substantial carbon emissions are traded internationally. Wiedmann et al. [67] used an MRIO model to calculate the CF of UK and disclosed that a widening emissions gap between producer and consumer increased the deficit in the balance of EET. In addition, Wiedmann et al. [54] expanded the application range of IOA to a material footprint as an indicator of resource use. The results showed that developed economies had made smaller achievements in relative or absolute decoupling than previously reported. Ivanova et al. [68] developed a CF inventory associated with household consumption for 177 regions in 27 EU countries and evaluated the driving forces through a set of technical, geographic, social, and economic factors. The results revealed that income was the most important driver of a region’s CF.

### 3.3. Analysis of Article Structure Variation

In this part, a structural variation analysis (SVA) is conducted to explore the information from citation links bridging distinct clusters and the trajectories of several contributors in the science mapping. The main purpose of SVA is to detect new types of remote connections or unprecedented intercluster bridges and to explain the novelty and value of these specific connections. According to the theory of SVA, modularity change rate indicates the extent to which an article changes the existing knowledge structure and reflects its transformative potential. The higher the modularity change rate is, the greater the transformative potential of an article. Thus, SVA provides a way to identify potential articles with extraordinary connections across distinct clusters, while the theories of scientific discovery have proved that many significant contributions result from boundary-spanning ideas [42].

#### 3.3.1. Trajectories of Prolific Authors

The trajectories of prolific authors in the landscape of different clusters reflects the leading contributions they have made in the research field. To analyze the trajectories of representative prolific authors, such as Lenzen and Wiedmann, who are ranked the highest two authors in terms of frequency (Table 1), is an effective approach to explore the hidden connections between citation links across distinct clusters.

(1)Manfred Lenzen is a prolific researcher with several seminal articles featured in different clusters (Figure 8b). In Figure 8, the dashed lines indicate novel co-citation links, and the stars indicate the articles that are both cited and citing. The citation trajectories of Manfred Lenzen create new dense and complex connections between clusters #0 carbon footprint, #1 urban transformation, #2 international trade, #5 input–output analysis, and #9 local consumption. He has contributed major methodological advances and applications to the areas of GHG emissions, life-cycle assessment, and IOA. In the last few years, he has been studying CF using IOA in the areas of the world economy, international trade, and environmental responsibility.(2)Thomas Wiedmann is another prolific researcher (Figure 8c). His citation trajectories move across the citation landscape among clusters #0 carbon footprint, #1 urban transformation, #2 international trade, #5 input–output analysis, #8 life-cycle assessment, and #9 local consumption. He is also a close coauthor of Lenzen. Wiedmann is committed to using the MRIO model to research global sustainability questions. He introduced the data and methodological and institutional requirements for MRIO analysis in detail and presented the framework of the international MRIO model, which was an important reference for subsequent research.

#### 3.3.2. Articles with Transformative Potential

Articles from well-known prolific authors tend to get more attention generally. However, newly published articles of potential scholars may be overlooked by the citation-based indicators. It is necessary to focus on the conceptual structure effects of newly published articles in the knowledge domain [69]. As Figure 8d–g shows, four articles with high modularity change rates and published in recent years were identified, making connections across distinct clusters with transformative potential.

(1)The appearance of “*Carbon and material footprints of a welfare state: Why and how governments should enhance green investments*” (Figure 8d) created a new bridge between clusters #0 carbon footprint, #3 household consumption, #4 household budget survey, and #7 material footprint. This article was published by Ottelin [70] in 2018. In this study, the authors examined how public spending affects material footprints and CFs, which had been neglected in previous literature. This study also revealed that income transfer and public welfare services can improve carbon equity among citizens.(2)The appearance of “*Developing a city-centric global multiregional input-output model (CCG-MRIO) to evaluate urban carbon footprints*” (Figure 8e) created a new bridge between clusters #0 carbon footprint, #1 urban transformation, and #4 household budget survey. This article was published by Lin [71] in 2017. In this study, the authors developed a city-centric GMRIO model to measure the CF at the city level, while before that, it generally lacked MRIO tables. The main contribution of this study is that it filled the gap in data and methods in urban consumption-based CF compiling, and opened a new scope for policymaking on the demand side.(3)The appearance of “*Exploring the material footprints of national electricity production scenarios until 2050: The case for Turkey and UK*” (Figure 8f) created a new bridge between clusters #0 carbon footprint, #4 household budget survey, #7 material footprint, and #11 sustainable manufacturing. This article was published by Kucukvar [72] in 2017. In this study, the authors built a GMRIO model to analyze the material footprint of electricity production, which combined with three energy-production scenarios in Turkey and the UK. As a vital development for the MRIO model, this study was the first in-depth research of the material footprint of mineral resources.(4)The appearance of “*Trade and the role of non-food commodities for global eutrophication*” (Figure 8g) created a new bridge between clusters #0 carbon footprint, #3 household consumption, and #4 household budget survey. This article was published by Hamilton [73] in 2018. In this study, the authors focused on the environmental impacts of non-food commodities and estimated global marine and freshwater eutrophication footprint occurring along the global supply chains with the MRIO model. This study expanded the application field of the MRIO model from CFs to eutrophication footprint and provided a better understanding of the role of non-food commodities in driving eutrophication impacts.

### 3.4. Analysis of Keywords Co-Occurrence Network

In this part, co-occurrence network analysis is discussed to explore research hotspots and trends. A series of parameters in CiteSpace are set as follows: the time slicing is one year per slice and the data selection criterion is a modified g-index. After obtaining the keyword co-occurrence network, we clustered the keywords and extracted the noun terms as cluster names with a log-likelihood ratio (LLR) algorithm. The results present a total of 357 terms with 9 clusters and 1292 links.

#### 3.4.1. Analysis of High-Frequency Keywords and Clusters

The high-frequency keywords and co-occurrence network describe the outline of main contents in a research field. Table 5 demonstrates the top 20 high-frequency keywords that appeared in the studies of CF based on IOA approach over the period of 2008–2021: carbon footprint (159), consumption (148), input–output analysis (130), international trade (99), emission (86), impact (82), CO_2_ emission (80), greenhouse gas emission (77), energy (76), life-cycle assessment (66), trade (59), model (54), environmental impact (53), China (43), footprint (40), system (37), input–output (32), city (30), energy consumption (25), and policy (25). Each high-frequency keyword occurs at least 25 times, and most of them appear in the early stages of the research.

Figure 9 shows the keyword co-occurrence network in the time line. In the timeline visualization, clusters are displayed along horizontal timelines from left to right. Mean silhouette and modularity Q were 0.7416 (>0.5) and 0.4826 (>0.3), respectively, indicating that the clustering was reasonable and significant. The clusters, numbered from 0 to 8 on the right in Figure 9, are arranged vertically in descending order of their size [42]. Cluster #0 multiregion input–output analysis is the largest. The other clusters from large to small are #1 household consumption, #2 model, #3 water footprint, #4 LCA, #5 reliability, #6 urban households, and #7 system, #8 CO_2_. As shown in the time-line overview, clusters #0, #1, #2, #3, #4, and #8 cover a period of 14 years and remained active until 2021, but clusters #5 and #6 fade out after 2020. The clusters of keywords can represent the core content of existing studies. A further analysis of these clusters is needed to find some clues to the hotspots and trends in this research field.

#### 3.4.2. Analysis of Research Hotspots

Based on keyword co-occurrence network analysis, we can reveal the hotspots related to CF research based on IOA. The focus of those works varies with clusters. For example, cluster #0 multiregion input–output analysis and cluster #4 LCA represent research hotspots for model applications, while cluster #1 household consumption and cluster #6 urban households represent research hotspots for scales. To each cluster of the keyword co-occurrence network, we conducted an in-depth analysis as follows to display hotspot topics and topic distribution.

Cluster #0 focuses on multiregion input–output analysis, with an average publication year of 2015. MRIO is the most popular analysis method in the research on CF. This cluster represents extensive concern about the CF in cross-regional trade. The construction of IOT covering more countries and regions with more detailed and reliable department classification is very important for CF accounting with IOA. With the continuous improvement in databases, such as EORA, WIOD, Exiobase, and GTAP, MRIO has become the most effective IOA model.

Cluster #1 focuses on the impact of “household consumption” on CF, with an average publication year of 2014. With the popularity of the concept of consumer responsibility, CF associated with the consumption of services and goods has attracted more and more attention. A country’s CF can be split into household (private) consumption, public consumption, and investment [74]. The environmental impact of household consumption has remained one of the hot topics in CF studies over the last year. From the macro perspective, the carbon emissions of households are a main component—around 70% [48]. Computing the CF of a household under the responsibility principle refers to the cumulative carbon the consumed goods’ supply chains have emitted, either directly or indirectly. The embodied carbon in international trade through the final products consumed and supply chains have reached almost a third of total GHG emissions around the world [56,75,76]. These are challenges for consumer CF accounting, i.e., to allocate the household consumption data of detailed projects such as food, transportation, clothing and medical treatment to the corresponding departments of IOT.

Cluster #2 focuses on “model” and has the most high-frequency words. The mean publication year of cluster #2 is 2010, indicating that it is a relatively mature research field. The accounting model is the basis of CF research. At the macro level, IOM has become the main analysis tool of CF accounting, and can comprehensively reflect the carbon emission relationships between various sectors of economic system. The main accounting models of CF include pure IO, hybrid, SRIO, semi-MRIO, MRIO model, and it has become more and more accurate with spatial scope from country scale to regional scale over the last few years.

Cluster #3 focuses on “water footprint,” with an average publication year of 2015. Water footprint and CF both belong to the footprint family [77]. With the continuous development in CF research based on IOA, relevant methods are gradually applied to the research of the water footprint, especially accounting for it in international trade and the global supply chain. Similar research also extends to the metal footprint.

Cluster #4 focuses on “life-cycle assessment (LCA)” analysis, with an average publication year for of 2015. LCA is another important method of CF accounting, which is often compared with IOA. LCA is a “bottom-up” process-based analysis method, and has become the most important CF accounting method at the micro level, especially at the food or non-food product scale. LCA is a “cradle-to-grave” measurement method that contains the whole process of GHG emissions: from raw-material mining, processing, storage, transportation, and usage to waste treatment. Due to the complex division of labor, it is hard for LCA to define the boundary at the macro level, while IOA can offset these types of calculation errors [11]. Since both the LCA and IOA have advantages and disadvantages, researchers combined LCA with IOA to develop the economic input–output life-cycle assessment (EIO-LCA) or hybrid life-cycle assessment, which has made the boundary much clearer, and improved the accuracy of organizational CF accounting to a certain extent. One such case is Wiedmann [78] using the hybrid LCA method to calculate the indirect GHG emissions of wind power in the UK.

Cluster #5 focuses on the “reliability” of the CF accounting model, and has an average publication year of 2015. The preparation of IOTs with more detailed and reliable classification of different departments, covering more countries and regions, is very important to the development of IOA. However, the release of MRIO’s and other model’s databases are compiled by some international organizations based on official data instead of by the government directly, leaving the data sources and compilation methods are different. Therefore, numerous studies have conducted consistency and reliability tests on databases [52,60,79].

Cluster #6 focuses on the environmental impact of carbon emissions from “urban households,” with an average publication year of 2016, and is a relatively new research field. Human settlements, especially cities, associated with most human activities, are main drivers of carbon emissions [80,81]. Several city-level studies estimated the CF of urban households with IOM and evaluated the driving forces through social, economic, and technical factors [60,68,82].

Cluster #7 focuses on the “system” boundary of CF assessment, with an average publication year of 2014. The carbon emission baseline of a region is highly dependent on the system boundaries for which they are calculated [83]. Among the CF accounting methods, input–output approaches can resolve the boundary selection problem related to the so-called truncation errors. In a global MRIO analysis, the system boundary generally includes the direct, indirect (regional), and indirect (global) CFs [84].

Cluster #8 focuses on the “climate change” caused by GHG, with an average publication year of 2013. This cluster was concerned mainly the influence of local climate actions on climate change. It is noteworthy that cluster #8 is small but relatively active, sustaining a period of 14 years from 2008 to 2021. Thus far, as an important tools to measure GHG emissions, CF has been widely concerned and mentioned while discussing climate change across academic circles, the government, and the public.

#### 3.4.3. Analysis of Research Trends

To find the research trends of a discipline, it is necessary to explore the evolution of research objects, tools, and themes. For the research field of CF assessment with IOA, the research objects are mainly different scales of regions, sectors, or individuals etc., and the research tools are mainly different models of SRIO, MRIO, or EIO-LCA etc. The research themes change over time, while the relatively new and valuable keywords “supply chain” and “driver factors” (Figure 9) provide clues for the exploration of research trends. Finally, according to the keyword co-occurrence network and specific clusters, we analyze the emerging trends of the research hotspots.

##### The Research Scales Tend to Be More Microscopic

As the issues of carbon emission studies tend to be more complicated and comprehensive, the research scale of CF assessment is becoming more and more microscopic from national to subnational to sector and even household level. IOA for CF accounting is widely applied in the global trading system, global production system [85], economic systems [86], and energy systems [87] at the level of national, subnational, region, city, and household. CF assessment at the microlevel is conducive to formulating more feasible emission-reduction policies. At the subnational level, there are applications of IO models to calculate the CF for small spatial areas, such as cities, in response to the increasing demand of information on regional and local CF. It is key for local CF accounting to combine information on global production activities with local consumption activities, despite facing some challenges. At the sector level, the application of IO models in calculating the sectoral CF provide an accurate picture of the carbon emissions associated with consumption and production activities [49]. At the household level, using IO models to estimate consumption patterns and lifestyle-related carbon emissions of households provides a new avenue for CF applications.

##### The Applications of Models Tend to Be More Detailed

With the continuous improvement in the database that IOA is based on, the IOT gets more detailed with reliable department classification and covers more countries and regions. At the macroscale, the MRIO model resolves the error caused by the production efficiency differences in the SRIO model, which enhances the accuracy, and it has been widely used to study the CF through international trade between countries. As further improved by nesting the “rest of world” region, the MRIO model forms the GMRIO, which enables a closed planetary boundary and allows assessment of the CF of regions and even cities across countries embodied in the trade [88]. At the microscale, EIO-LCA or hybrid LCA improves the accuracy of CF accounting at household level, which is one of the future research directions. However, CF accounting with IOA is still subject to some restrictions, such as the preparation of IOT and the fact that the basic database has not been fully institutionalized and standardized. Due to the differences in IOT and databases, CF accounting results for the same case can differ. The preparation of IOT and construction of databases play a key role in improving the quality of CF accounting, and need to attract sustained attention in related fields from theory and practical areas in the future.

##### Supply-Chain Analysis and Driver-Factor Analysis

As CF topics tend to become more and more detailed and diversified, global supply-chain analysis and driver-factor analysis of CF with IOA will probably become the main research directions in the future. CF in the supply chain provides a basis for exploring key emission links, optimizing industrial structure and quantifying emission-reduction responsibilities. For instance, a global supply-chain distribution analysis of electronic products can present an indirect contribution to the total CF of each country. Driver-factor analysis will be another research trend in the future. CF-driver factors vary according to carbon efficiency, final demands of global production, sociodemographic trends, and so on. Structural decomposition analysis usually is applied in the IOA to quantify the contribution of these drivers.

## 4. Conclusions and Limitation

### 4.1. Conclusions

In this study, we turned our attention to the emerging area of CF analysis with IOA from recent years. A comprehensive review of 491 studies retrieved from the WOS in the period of 2008 to 2021 was conducted with the bibliometric analysis method. Through information visualization, knowledge-network, and knowledge-evolution analysis, the research field of IOA-related CF gets clearer, which provides guidance for future research. Preliminarily, we obtain an overall result showing that after continuous exploration and cooperation among global scholars of different institutions, IOA has become a mainstream CF-assessment tool that has formed a complex knowledge-network structure in the process of expansion. With improvements in basic database precision and accuracy, this research field will see more meticulous and diversified scales, models, and contents. Finally, we combed all the research results and grouped them as follows.

(1)Through basic feature analysis of the published articles, such as quantitative trend, author, country (region), institution, and journal, it was found that the research field of CF analysis based on IOA has continued to increase over the last 14 years. The top three producing countries are China, the USA, and Australia. Richard Wood, Manfred Lenzen, and Thomas Wiedmann are the top three representative scholars in this field, and there is a certain degree of cooperation among them. The Norwegian University of Science Technology, Leiden University, and University of Sydney are the active scientific research institutions. It was also found that CF analysis with IOA is labeled as a field with interdisciplinary characteristics involving nature, geographic, economic, and social fields.(2)Through the co-citation network analysis based on the cited articles, it was found that the classic references with high frequency, burst, and betweenness were mainly divided into three categories: related literature reviews, database application introduction, and CF accounting. The application characteristics of the CF analysis based on the IOA has formed its own knowledge domain. In addition, through article-structure-variation analysis, we identified four articles with high modularity change rates, making connections across distinct clusters with transformative potential. These important articles may change the direction of the research field in the future.(3)Through the keyword co-occurrence network analysis, we explored the hotspots and trends of the knowledge domain. In this research field, the top 20 high-frequency keywords were CF, consumption, input output analysis, international trade, emission, impact, CO_2_ emission, greenhouse, gas emission, energy, life-cycle assessment, trade, model, environmental impact, China, footprint, system, input–output, city, energy consumption, and policy. The keywords formed nine clusters numbered from 0 to 8: cluster #0 multiregion input–output, #1 household consumption, #2 model, #3 water footprint, #4 LCA, #5 reliability, #6 urban households, #7 system, and #8 CO_2_. Through analysis of research trends, it was found that the research scales tended to be more microscopic and applications of models more detailed. Supply-chain analysis and driver-factor analysis will probably become the main research directions in the future.

Based on all the conclusions of this research, we propose three pieces of advice to policymakers and environmental scientists. Firstly, strengthen the construction and improvement of the database. A detailed and accurate database is the basis and the premise of IOA research, and deserves more effort on improvement from policymakers and environmental scientists. Secondly, strengthen international cooperation on carbon emission reduction. The shared responsibility between countries around the world for reducing carbon emissions is a focal point. IOA can transform economic relationships into physical relationships and reflect the direct and indirect emission relationships between sectors or regions, which is significant for clarifying shared responsibility and improving intergovernmental cooperation in dealing with climate change. Thirdly, strengthen the integration of LCA and IOA tools. Policymakers and environmental scientists should pay more attention to EIO-LCA or hybrid LCA application for CF accounting of individuals and households, and provide better guidance for individuals to fulfill their carbon emission-reduction responsibilities.

### 4.2. Limitations

Scientometric visualization analysis is applied in this paper to fill the research gap in respects of the IOA model applied to CF studies. Although it is helpful to provide a better understanding of the basic feature, knowledge structure, and frontier trends in this field, there are still some limitations. (1) The scope of the data was limited by the topic-search combinations the retrieval sources, which led to insufficient coverage of the 491 selected literatures. (2) Although scientometric visualization analysis can provide a quick overview of the knowledge domain, the results vary with the selection of parameters, which might weaken the robustness of the results. (3) The summary of the co-citation network based on the focus topics, research hotspots, and research trends is classified in terms of relative parameter values and subjective judgment, which might not be comprehensive and lack objectivity. We will make tracks for following international science with scientometric visualization analysis and conduct more in-depth discussion in the future, thus providing a more accurate reference in this research field.

## Figures and Tables

**Figure 1 ijerph-19-11343-f001:**
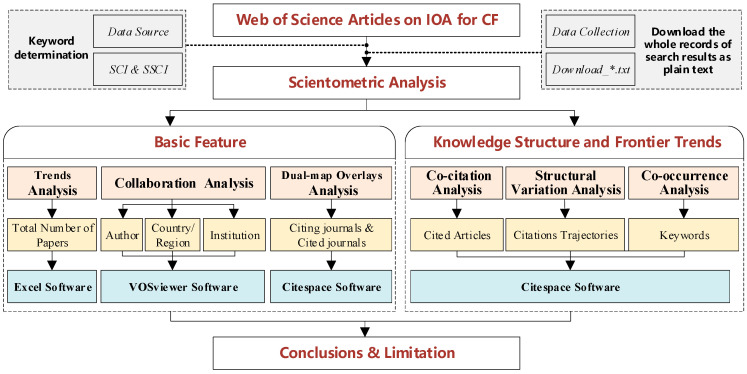
Outline of research design.

**Figure 2 ijerph-19-11343-f002:**
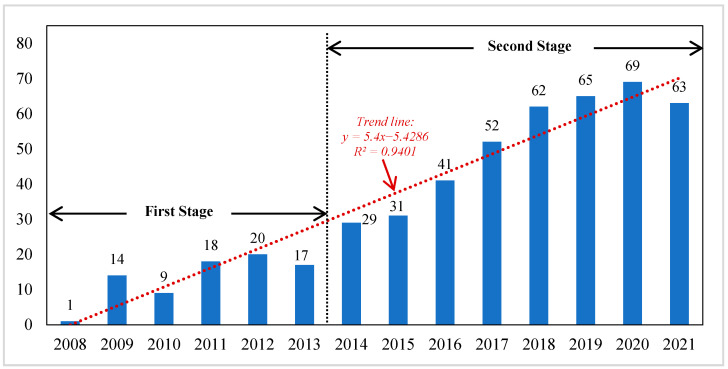
The number of published papers on CF with IOA approach (2008–2021).

**Figure 3 ijerph-19-11343-f003:**
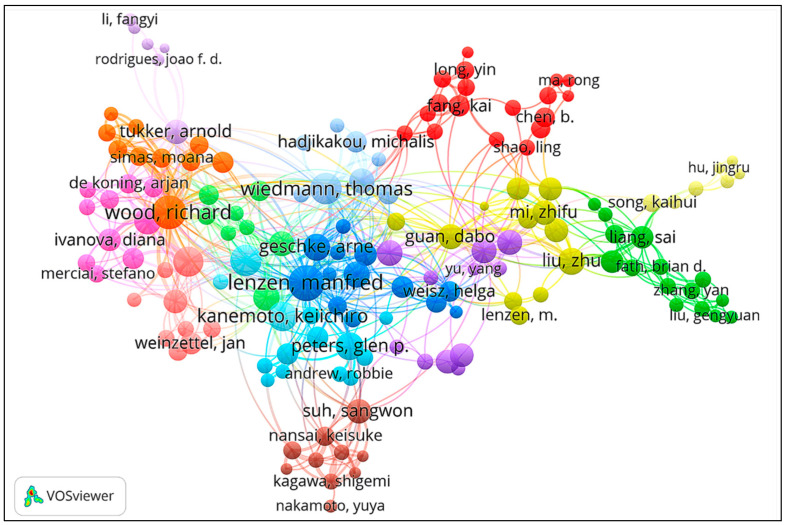
Network map of authors collaboration.

**Figure 4 ijerph-19-11343-f004:**
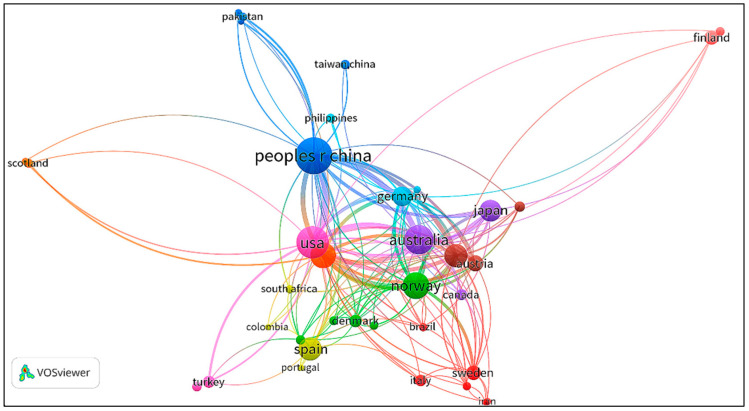
Network map of country (region) collaboration.

**Figure 5 ijerph-19-11343-f005:**
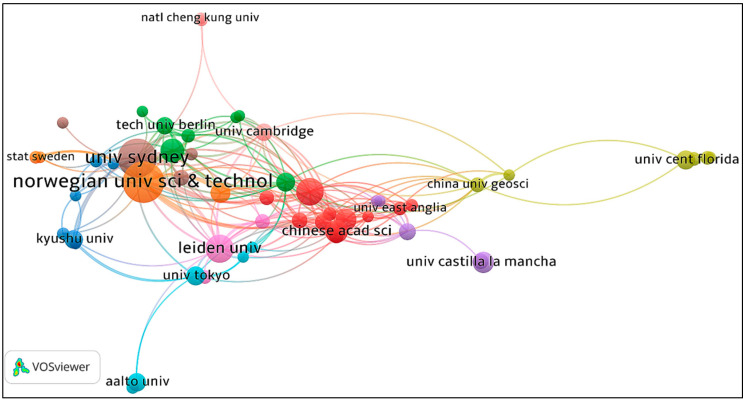
Network map of institution collaboration.

**Figure 6 ijerph-19-11343-f006:**
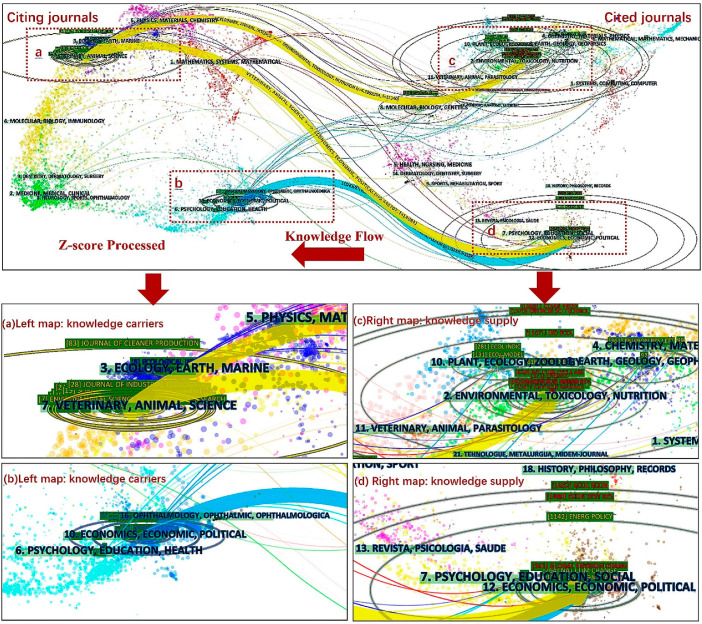
Dual-map overlay graph of journals and disciplines after Z-score processing(**top**); detail in enlarged scale (**bottom**): knowledge carriers (**a**,**b**), knowledge supply (**c**,**d**).

**Figure 7 ijerph-19-11343-f007:**
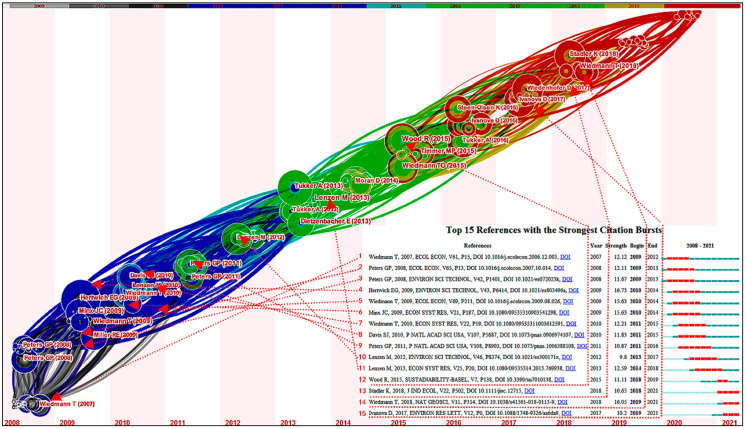
Time-zone view of the cited articles.

**Figure 8 ijerph-19-11343-f008:**
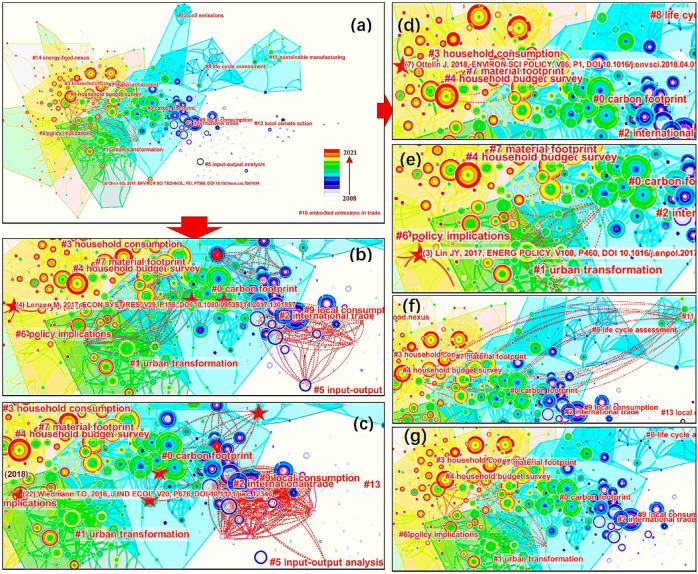
Whole clusters of cited articles. (**a**); Novel co-citations of 6 papers by Lenzen (**b**) and 7 papers by Wiedmann (**c**). Novel co-citations of papers by Ottelin (**d**), Lin (**e**), Kucukvar (**f**), and Hamilton (**g**).

**Figure 9 ijerph-19-11343-f009:**
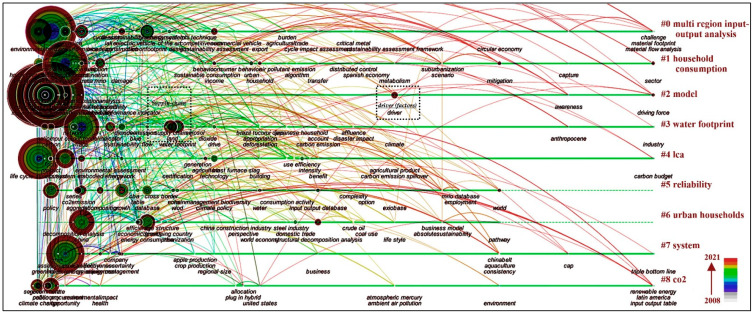
Time-line visualization of the keyword co-occurrence network.

**Table 1 ijerph-19-11343-t001:** Top 10 authors based on frequency.

Author	Frequency	Percentage	Author	Frequency	Percentage
Wood Richard	36	7.3%	Moran Daniel	16	3.3%
Lenzen Manfred	29	5.9%	Tukker Arnold	16	3.3%
Wiedmann Thomas	26	5.3%	Onat Nuri Cihat	12	2.4%
Hertwich Edgar G	21	4.3%	Peters Glen	12	2.4%
Kucukvar Murat	19	3.9%	Tatari Omer	12	2.4%

**Table 2 ijerph-19-11343-t002:** Top 10 countries (regions) based on frequency.

Country (Region)	Frequency	Percentage	Country (Region)	Frequency	Percentage
China	142	28.9%	Netherlands	50	10.2%
USA	102	20.8%	Spain	48	9.8%
Australia	82	16.7%	Japan	44	9.0%
Norway	68	13.8%	Germany	34	6.9%
England	64	13.0%	Austria	22	4.5%

**Table 3 ijerph-19-11343-t003:** Top 10 institutions based on frequency.

Institution (First Author)	Frequency	Percentage
Norwegian University of Science and Technology	56	11.4%
Leiden University	54	11.0%
University of Sydney	51	10.4%
Beijing Normal University	28	5.7%
University of New South Wales Sydney	28	5.7%
Chinese Academy of Sciences	22	4.5%
University of Leeds	21	4.3%
Tsinghua University	17	3.5%
Universidad De Castilla La Mancha	17	3.5%
Yale University	17	3.5%

**Table 4 ijerph-19-11343-t004:** The top 10 cited articles with high co-citation frequency and betweenness centrality.

Title	Year	Author	Source	Freq	Centrality	Burst
*Carbon footprint of nations: a global, trade-linked analysis*	2009	Edgar G. Hertwich [48]	*Environmental Science and Technology*	44	0.11	16.74
*Input–Output analysis and carbon footprinting: an overview of applications*	2009	J.C. Minx [49]	*Economic Systems Research*	38	0.06	15.64
*A review of recent multi-region input–output models used for consumption-based emission and resource accounting*	2009	Thomas Wiedmann [50]	*Ecological Economics*	38	0.02	15.64
*Growth in emission transfers* via *international trade from 1990 to 2008*	2011	Glen P. Peters [51]	*Proceedings of the National Academy of Sciences of the United States of America*	39	0.02	16.08
*Building eora: a global multi-region input–output database at high country and sector resolution*	2013	Manfred Lenzen [52]	*Economic Systems Research*	49	0.05	12.53
*Global multiregional input–output frameworks: an introduction and outlook*	2013	Arnold Tukker [53]	*Economic Systems Research*	43	0.13	8.86
*The material footprint of nations*	2013	Thomas Wiedmann [54]	*Proceedings of the National Academy of Sciences of the United States of America*	40	0.07	9.09
*Global sustainability accounting—developing exiobase for multi-regional footprint analysis*	2015	Richard Wood [55]	*Economic Systems Research*	49	0.07	11.09
*Environmental and social footprints of international trade*	2018	Thomas Wiedmann [56]	*Nature Geoscience*	39	0.09	10.89
*Exiobase 3: developing a time series of detailed environmentally extended multi-regional input-output tables*	2018	Konstantin Stadler [57]	*Journal of Industrial Ecology*	36	0.01	10.64

**Table 5 ijerph-19-11343-t005:** The top 20 high-frequency keywords.

Keyword	Freq	Centrality	Year	Keyword	Freq	Centrality	Year
carbon footprint	159	0.04	2009	trade	59	0.03	2011
consumption	148	0.01	2009	model	54	0.06	2009
input–output analysis	130	0.02	2008	environmental impact	53	0.09	2008
international trade	99	0.07	2008	China	43	0.05	2011
emission	86	0.07	2008	footprint	40	0.1	2009
impact	82	0.13	2010	system	37	0.07	2010
CO_2_ emission	80	0.11	2009	input output	32	0.1	2009
greenhouse gas emission	77	0.09	2010	city	30	0.14	2009
energy	76	0.11	2009	energy consumption	25	0.04	2014
Life-cycle assessment	66	0.05	2008	climate change	25	0.13	2008

## Data Availability

Not applicable.

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
