# Peer review of "Carbon Footprint Research Based on Input–Output Model—A Global Scientometric Visualization Analysis"

_ijerph, 2022, doi:10.3390/ijerph191811343_

Round 1
Reviewer 1 Report
It is an honor for me to review an MS entitled; “Carbon footprint research based on Input-output Model – A global scientometric visualization analysis”. The author did an extraordinary effort but few useful suggestions can improve the research write up. Following points can be considered for the improvement of the MS before its acceptance for the publication;
· First of all, I suggest to give the outline of the MS.
· Try to write the full form of any (e.g., carbon footprint (CF)) abbreviation for the first time in the MS and use only abbreviations (e.g., CF) rather than full form in rest of the MS. Try to use a single format for all abbreviations in complete article as you haven’t followed in the complete MS.
· Try to write the full form of CO2 (e.g., carbon dioxide) as well for the first time in the MS (see introduction part).
· You have written a statement “Scholars from 51 countries (regions) carried out related research” in result and discussion part but you didn’t give the names of those 51 countries in any table. Please add another table of 51 countries or remove this statement from MS (Unnecessary).
· In the part of 3.1.3. Dual-map overlays; you have written “engineering environmental, engineering chemical” correct the write up order likewise; environmental engineering, chemical engineering. Write the full form of JCR Journal (abbreviation).
· Unable to read and understand the texts given into Fig. 6 and 10 due to very dark colours combination, texts are mixing up in different colours. Enlarge the text size in all figures. It should be easily readable
· Try to start a sentence or paragraph with the general word instead of the scientist's name (e.g. Tukker et al. (2009), Lenzen et al. (2012; 2013), Dietzenbacher et al. (2013), Peters et al. (2011)
· In Table. 4 please give the publications detail in a year-wise proper order (like; 2009, 2011, ….)
· Line 28-29: Fig.9 shows, four articles with high modularity change rates and published in recent five years (mentioned years as per Ottelin, Lin, Kucukvar and Hamilton in fig.9 as well???)
· Try to add few latest references of 2022 publications into the MS.
· Set all the references according to the Journal requirement because references are not written as per Int. J. Environ. Res. Public Health format and cross match the references as well carefully.
Thanks
Author Response
Dear reviewer:
Great thanks for your professional review work on our manuscript. The comments are all valuable and very helpful for revising and improving our paper. We have carefully considered the suggestion and make some changes which we hope meet with approval. The point-by-point response has been uploaded as a Word. Please see the attachment.

Reviewer 2 Report
This paper provides an innovative and useful analysis of th carbon footprinting literature. The main criticisms I have are just that the writing needs to be edited and that the figures need to be made clearer. Esp Fig 6., Fig 8, 10 But really all the figures would be greatly improved. If the current figures appear to be the direct output from the two software packages then I feel that the authors should take those images and have them re-done by a graphic artist to clearly illustrate the points being made. Some of the figures are really of limited value (Fig 9) as they are and could be removed.
Author Response

(The authors gave the same response as above.)

Reviewer 3 Report
Reviewer report
Carbon footprint research based on Input-output Model – A global scientometric visualization analysis
Q1) Carbon footprint (CF), is widely used to deal with the threat of climate change
Carbon footprint has been used recently. It was CO2 emissions that were commonly used for climate change. Revise your statement.
Q2) Ree, 1992; Wackernagel & Ree, 1996). Ree is misspelled. It will be Rees. Please check.
Q3) Instead of chapter, section might be a more appropriate expression. This is not a book, it is an article.
Q4) Your introduction is insufficient. The purpose of your article and why such an article is needed should be explained in more detail. More information on the carbon footprint should be presented. You can benefit from the following article and more.
https://doi.org/10.3390/w13101387
Q5) The conclusion part should be developed. What is the overall result about the carbon footprint of the 491 articles reviewed as a result of your research? How should policymakers and environmental scientists follow up on these results?
The work is generally well designed and of high quality. My decision is minor revision.
Author Response

(The authors gave the same response as above.)

Reviewer 4 Report
This manuscript delivers a preliminary review of the global carbon footprint researches by concentrating on the Input-output Model. However, further improvement is reqiured to display the incremental contributions of the reseearch. A couple of recommendations are as follows:
1) to clearly indicate the current knowledge gaps in this field by collecting more recent papers including the Global research on carbon footprint: A scientometric review, published in the EIA review in 2021.
2) to develop a more academic analysis on IOM in carbon footprint studies besides the existing primary illustrations based on figures and tables, and to make more approriate recommendations on further researches in this field.
Author Response

(The authors gave the same response as above.)

Round 2
Reviewer 4 Report
The revisied manuscript is fine.